# Visceral Obesity and Lipid Profiles in Chinese Adults with Normal and High Body Mass Index

**DOI:** 10.3390/diagnostics12102522

**Published:** 2022-10-17

**Authors:** Yutong Lu, Na Li, Tamotsu Kamishima, Peng Jia, Dan Zhou, Karen Hind, Kenneth Sutherland, Xiaoguang Cheng

**Affiliations:** 1Graduate School of Health Sciences, Hokkaido University, North-12 West-5, Kita-ku, Sapporo 060-0812, Japan; 2Department of Radiology, Beijing Jishuitan Hospital, Beijing 100035, China; 3Faculty of Health Sciences, Hokkaido University, North-12 West-5, Kita-ku, Sapporo 060-0812, Japan; 4Department of Radiology, BenQ Medical Center, The Affiliated BenQ Hospital of Nanjing Medical University, Nanjing 210019, China; 5Wolfson Research Institute for Health and Well-Being, Durham University, Durham DH1 3LE, UK; 6Global Center for Biomedical Science and Engineering, Hokkaido University, North-15 West-7, Kita-ku, Sapporo 060-8638, Japan

**Keywords:** body mass index, cardiovascular risk, lipid profile, obesity, visceral fat

## Abstract

Background: This study examined the prevalence of visceral obesity in Chinese adults across different body mass index (BMI) groups and their associated lipid profiles and demographic risk factors. Methods: A total of 1653 Chinese adults were recruited for the study. Abdominal quantitative computed tomography (CT) imaging was performed to derive the visceral adipose tissue (VAT) at the lumbar vertebrae (L2–L3) levels. Visceral obesity was defined using established cutoff values. Fasting serum total cholesterol, total glucose, high-density lipoprotein, and low-density lipoprotein were measured. Results: Visceral obesity was prevalent in 35% of men and 22% of women with normal BMI (18.5–24 kg/m^2^) and 86% of men and 78% of women with high BMI (≥24 kg/m^2^). In both sexes, participants with normal BMI and visceral obesity had higher levels of TC, TG and LDL and lower HDL compared to those with normal VAT. The risk factors for visceral obesity in women with normal BMI were an age ≥50 years and BMI ≥22.3 kg/m^2^ and in men included a BMI ≥22.5 kg/m^2^. Conclusion: Visceral obesity was observed in the participants with normal BMI and was associated with an adverse lipid profile. The BMI cutoff points were lower than the normally accepted values.

## 1. Introduction

The prevalence of obesity across Asia has increased rapidly over the last two decades. This has become a major health challenge, given the associated elevated risk of disease, including type-2 diabetes mellitus, hypertension, metabolic syndrome, and an increased risk of mortality [1,2]. Obesity is defined by the World Health Organization (WHO) as excessive fat accumulation, which is a detriment to health. It is diagnosed as a body mass index (BMI) of ≥30 kg/m^2^. About 13% of the world’s adult population (11% of men and 15% of women) were obese in 2016 [3]. However, while BMI is generally considered the best measure for assessing obesity, those patients with only central obesity may not be considered at an increased risk of obesity-related cardiovascular disease risk factors [4]. Accumulating evidence suggests that, for any given BMI, Chinese adults have a greater percentage of body fat and a higher risk of metabolic syndrome [5]. On this basis, the Working Group on Obesity in China (WGOC) recommends using a BMI cutoff of 24 kg/m^2^ for overweight and 28 kg/m^2^ for obesity [6].

There is evidence for the existence of a population sub-group who, despite having normal BMI, display clusters of risk factors for metabolic and cardiovascular disease, such as insulin resistance, dyslipidemia, and increased visceral adipose tissue (VAT) [7]. VAT is important because it is associated with the increased circulation of pro-inflammatory cytokines and oxidative stress, which may lead to the diseases mentioned above [8,9]. Accumulating research also indicates that visceral fat is an independent predictor of the components of metabolic syndrome [10], leading to cardiovascular disease. Visceral fat is also an independent predictor of all-cause mortality in men and women [11]. Although quantifying the total visceral adiposity in an individual is time-consuming, a recent study supports the use of quantitative computed tomography (QCT) for the measurement of abdominal VAT at the umbilical level and at the level of the lumbar vertebrae (L2–L3) [12]. In the Chinese population, VAT scores of ≥142 cm^2^ in men and >115 cm^2^ in women are identified as the cutoff points for visceral obesity and have been associated with a higher prevalence of cardiovascular disease risk factors, including hypertension, elevated low high-density lipoprotein (HDL) and total cholesterol (TC), and/or hypertriglyceridemia and hyperglycemia [13]. In overweight and obese Chinese adults, high visceral fat measured by CT has been associated with an adverse lipid and glucose profile [14]. However, in adults with normal BMI, visceral obesity and other early signs of cardiometabolic disease may go undetected, because these individuals are not usually be referred for cardiometabolic screening. The purpose of the present study was to explore the prevalence and phenotypic predictors of visceral obesity in Chinese adults with normal BMI and high BMI and to explore their corresponding lipid profiles.

## 2. Materials and Methods

### 2.1. Study Population

The study was approved by the ethics committee of Beijing Jishuitan Hospital and the Ethics Committee of Nanjing BENQ Medical Center. Informed consent was obtained from all subjects involved in the study. A total of 1813 Chinese adults from Nanjing who underwent regular health checks at Nanjing BEQN Medical Center from January 2013 to December 2016 were included in this study. From this number, 132 participants were excluded due to invalid BMI or visceral fat data (height and/or weight were not collected, or VAT was not measured) (Figure 1), and 28 were excluded because their BMI was <18.5 kg/m^2^. We therefore analyzed a total of 1653 participants (age range, 21–80 years old; average, 47.94 years old; BMI range, 18.51–37.76 kg/m^2^, average 25.24 kg/m^2^), including 1044 men and 609 women. The exclusion criteria were (i) current treatment with systemic corticosteroids, (ii) cirrhosis with ascites, (iii) known hyperthyroidism or hypothyroidism, (iv) cancer, (v) severe disability, (vi) psychiatric disturbance, or (vii) computed tomography (CT) contraindications, such as pregnancy.

### 2.2. Anthropometric Measurements

Height (cm) and weight (kg) were measured by a staff member of the medical examination center using a free-standing stadiometer and electronic scales (Henan Shengyuan Industry Ltd. Henan, China). Participants took off their shoes and wore comfortable clothes during the measurements. BMI was calculated as weight (kg) ÷ height ^2^ (m). According to the WGOC, a BMI over 24 kg/m^2^ was considered high. Body surface area (BSA) was calculated as 0.0061 × height (cm) + 0.0128 × weight (kg) − 0.1529 [15].

### 2.3. Visceral Fat Measurements

The VAT area at the L2–L3 level was measured using a 64-detector row computed tomography (CT) scanner (Light Speed VCT Vision; GE Healthcare, Milwaukee, WI, USA), operated by a qualified radiologist. The parameters of CT were 120 kV and 100 mAs, the slice thickness was 1 mm, and the field of view was 40 cm. The original CT data in QCT format were imported into the Tissue Composition Module Beta 1.0 (Mindways, Austin, TX, USA), and adipose tissue was segmented and mapped in blue with default thresholds by two trained radiologists (one measured, one reviewed). The VAT was semi-automatically measured by the software, and the final results were exported to Excel.

The CT instrument was air-corrected every day when the machine was turned on. A function inspection was conducted once a quarter by the hospital equipment engineer, including a power check, alarm check, accessories check, time check, battery check, and parameter check, and these checks were performed annually by the engineer of the equipment manufacturer.

### 2.4. Blood Lipid Measurements

Fasting blood samples were collected for the measurement of TC, TG, LDL, and HDL by immunoturbidimetry with the Roche COBAS8000 Biochemical luminescence assembly line (Roche Diagnostics, Mannheim, Germany). All samples were collected and stored at room temperature and centrifuged for two hours. The collected samples were refrigerated for seven days at 2 to 8 degrees Celsius for review to mitigate any doubt.

### 2.5. Statistical Analysis

Data were analyzed using SPSS version 19.0 (SPSS Inc., IBM, Armonk, NY, USA) for Windows. Participants were grouped according to BMI using the Chinese standard (normal BMI: 18.5–24 kg/m^2^, *n* = 609, 36.2%, men = 290, women = 319; high BMI: ≥24 kg/m^2^, *n* = 1044, 84.0%, men = 754, women = 290) and by VAT based on the established cutoff points for visceral obesity in the Chinese population (men ≥142 cm^2^; women ≥115 cm^2^) or for normal VAT (men < 142 cm^2^; women
<115 cm^2^) [13]. Descriptive results were expressed as the mean ± standard deviation. The chi-square test was used to determine whether there was a difference in the prevalence of visceral obesity between men and women with normal BMI. Independent t-tests were used to compare the descriptive variables (age, BMI, and BSA) and lipid profiles between the visceral obesity and normal-VAT sub-groups. The collinearity diagnoses in the linear regression analysis were used to evaluate multiple linear relationships between variables. There was no evidence of multicollinearity between age, BMI, or BSA in the normal BMI group (men: variance inflation factor (VIF) = 1.12, 1.28, and 1.41, respectively; women: VIF = 1.23, 1.34, and 1.39, respectively) and the high BMI group (men: VIF = 1.12, 1.62, and 1.77, respectively; women: VIF = 1.23, 1.62, and 1.79, respectively). The area under the curve (AUC) and its 95% confidence interval (CI) were obtained by analyzing the receiver operating characteristic (ROC) of the variable, and the cutoff value was calculated on the basis of grouping the variable that predicted a high or normal VAT. Logistic regression analysis was used to evaluate the relationship between each binary variable and VAT. Odds ratios (ORs) with 95% CIs were extrapolated. Significance was defined as *p* < 0.05.

## 3. Results

Table 1 and Table 2 present the descriptive results for men and women, respectively, according to the BMI and VAT sub-groups. Visceral obesity was prevalent in 101 men (34.8%) and 70 women (21.9%) with normal BMI and 651 men (86.3%) and 226 women (77.9%) with high BMI. Among people with a normal BMI, men were more likely than women to have a VAT above the cutoff value. The difference was statistically significant (χ^2^ = 12.5, *p* < 0.001). In men with normal BMI, those with high VAT had greater BMI and BSA scores (*p* < 0.001). Women with normal BMI and high VAT were older (*p* < 0.001) and had greater BMI (*p* < 0.001) and BSA (*p* = 0.013) scores compared to those with normal VAT.

In men with normal BMI, those with high VAT had greater BMI and BSA scores (*p* < 0.001). Women with normal BMI and high VAT were older (*p* < 0.001) and had greater BMI (*p* < 0.001) and BSA (*p* = 0.013) scores compared to those with normal VAT.

### 3.1. Phenotypic Predictors of Visceral Obesity

Figure 2 and Figure 3 present the ROC analysis of BSA and BMI as predictors of visceral obesity in normal BMI men and women, respectively, while Figure 4 and Figure 5 present the ROC analysis of BSA and BMI as predictors of visceral obesity in high BMI men and women, respectively. The thresholds of BSA and BMI were used as criteria for the multivariate logistic regression analysis grouping.

In normal-weight men, the AUC for BSA was 0.626 (95% CI = 0.559–0.693; *p* < 0.001), and for BMI it was 0.719 (95% CI = 0.659–0.779; *p* < 0.001). The optimal cutoff value for BSA was 1.72 m^2^, with the highest sensitivity (59.4%) and specificity (60.8%), while the optimal cutoff value for BMI was 22.5 kg/m^2^ (sensitivity, 74.3%; specificity, 63%). For men with high BMI, the AUC for BSA was 0.666 (95% CI = 0.611–0.721; *p* < 0.001), and for BMI it was 0.750 (95% CI = 0.701–0.799; *p* < 0.001). The optimal cutoff value for BSA was 1.87 m^2^, with the highest sensitivity (60.2%) and specificity (68.0%), while the optimal cutoff value for BMI was 26.3 kg/m^2^ (sensitivity, 66.2%; specificity, 72.8%).

In normal-weight women, the AUC for BSA was 0.602 (95% CI = 0.526–0.678; *p* < 0.001), and for BMI it was 0.746 (95% CI = 0.685–0.808; *p* < 0.001). The optimal cutoff values were 1.57 m^2^ for BSA and 22.3 kg/m^2^ for BMI (sensitivity, specificity; 41.4%, 77.9%; 75.7%, 69.1%). In high-BMI women, the AUC for BSA was 0.628 (95% CI = 0.553–0.703; *p* = 0.002), and for BMI it was 0.731 (95% CI = 0.664–0.799; *p* < 0.001). The optimal cutoff values were 1.62 m^2^ for BSA and 25.3 kg/m^2^ for BMI (sensitivity, specificity; 69.0%, 57.8%; 75.7%, 62.5%).

Table 3 presents the results of the multivariate logistic regression analysis of the normal BMI groups. In men, only a BMI > 22.5 kg/m^2^ (OR 4.41, 95% CI 2.52–7.69; *p* < 0.001) independently predicted visceral obesity, while for women, an age of ≥ 50 years (OR 5.43, 95% CI 2.82–10.5; *p* < 0.001) and BMI >22.3 kg/m^2^ (OR 5.18, 95% CI 2.71–9.89; *p* < 0.001) independently predicted visceral obesity. These models correctly classified 69.3% of male and 81.2% of female participants.

In men, only a BMI > 22.5 kg/m^2^ (OR 4.41, 95% CI 2.52–7.69; *p* < 0.001) independently predicted visceral obesity, while for women, an age of ≥ 50 years (OR 5.43, 95%CI 2.82–10.5; *p* < 0.001) and BMI > 22.3 kg/m^2^ (OR 5.18, 95% CI 2.71–9.89; *p* < 0.001) independently predicted visceral obesity. 

Table 4 shows the results of the high BMI groups. In men, a BSA > 1.72 m^2^ (OR 1.84, 95% CI 1.14–2.96; *p* = 0.012) and BMI > 26.4 kg/m^2^ (OR 4.26, 95% CI 2.61–6.96; *p* < 0.001) independently predicted visceral obesity, while for women, an age of ≥50 years (OR 3.77, 95%CI 1.94–7.31; *p* < 0.001), BSA > 1.62 m^2^ (OR 2.99, 95%CI 1.53–5.85; *p* = 0.001), and BMI > 25.3 kg/m^2^ (OR 3.99, 95% CI 2.13–7.46; *p* < 0.001) independently predicted visceral obesity. These models correctly classified 86.3% of male and 79.9% of female participants.

In men, a BSA > 1.72 m^2^ (OR 1.84, 95% CI 1.14–2.96; *p* = 0.012) and BMI > 26.3 kg/m^2^ (OR 4.26, 95% CI 2.61–6.96; *p* < 0.001) independently predicted visceral obesity, while for women, an age of ≥ 50 years (OR 3.77, 95%CI 1.94–7.31; *p* <0.001), BSA > 1.62 m^2^ (OR 2.99, 95%CI 1.53–5.85; *p* = 0.001), and BMI > 25.3 kg/m^2^ (OR 3.99, 95% CI 2.13–7.46; *p* < 0.001) independently predicted visceral obesity. 

### 3.2. Lipid Profiles of All Participations

Among the 1653 participants, 624 (37.8%) and 619 (37.5%) people had TC and TG values higher than normal (TC ≥ 5.18 mmol/L and TG ≥ 1.7 mmol/L [16]), respectively, while 465 (28.1%) had LDL levels higher than the cutoff value (LDL < 3.37 mmol/L [16]) and 47 (21%) had HDL levels lower than normal (HDL ≥ 1.04 mmol/L [16]). Among the 1044 men, the number of abnormal values of TC, TG, LDL, and HDL mentioned above were 396 (37.9%), 478 (45.8%), 306 (29.3%), 298 (28.5%), respectively, and among 609 women, they were 228 (37.4%), 141 (23.2%), 159 (26.1%), 49 (8.1%), respectively.

### 3.3. Lipid Profiles of High vs. Normal Visceral Fat Sub-Groups

Table 5 and Table 6 present the fasting serum lipid profiles according to the BMI and VAT sub-groups for men and women, respectively. Men and women with normal BMI and high VAT had significantly greater TC, TG, and LDL and lower HDL than those with normal VAT. In both sexes, those with high BMI and high VAT had greater levels of TG and lower HDL.

Men and women with normal BMI and high VAT had significantly greater TC, TG, and LDL and lower HDL than those with normal VAT. In both sexes, those with high BMI and high VAT had greater levels of TG and lower HDL.

In high-VAT men, the mean levels of TG were above the reference threshold for dyslipidemia (TG ≥ 1.7 mmol/L). In the normal-BMI, high-VAT women, both TC and TG were above the reference thresholds for dyslipidemia (TC ≥ 5.18 mmol/L and TG ≥ 1.7 mmol/L). In the high-BMI, high-VAT women, TG was above the reference threshold. In both men and women, the values of LDL, HDL, and LDL/HDL in the high VAT and the low VAT groups were within the normal range (LDL < 3.37 mmol/L; HDL ≥ 1.04 mmol/L; LDL/HDL > 5 [17,18]).

## 4. Discussion

In this study, about one-third of the participants had dyslipidemia, similar to a previous report giving a value of about 41.9% of Chinese adults [19]. At the same time, we found that Chinese adults with normal BMI were more likely to be viscerally obese if their BMI was in the upper range of normal, and the risk was further increased in women aged 50 years or over. In addition, our findings also indicate the clinical relevance of visceral adiposity in individuals with normal BMI, given the accompanying higher fasting levels of serum TC, TG, and LDL, which imply an increased risk for the development of metabolic syndrome in this subpopulation.

The increasing prevalence of metabolic disease in Asian populations is more closely related to body fat distribution than to BMI [20]. In particular, visceral adipose tissue secretes adipocytokines, which are implicated in the development of metabolic syndrome and cardiovascular disease. Indeed, numerous studies have reported associations between abdominal fat and insulin resistance [21,22]. Visceral obesity constitutes the abnormally high deposition of visceral fat and is an independent component of metabolic syndrome [23]. In normal-weight Chinese adults, a higher waist circumference has been associated with a greater prevalence of insulin resistance and indicators of metabolic syndrome [22]. To our knowledge, this study is the first to report the prevalence of CT-defined visceral obesity in Chinese adults with normal BMI, with the associated lipid and glucose profiles. We found that adults with normal BMI and excess visceral fat had a more adverse lipid profile that was non-disinguishable from the lipid profiles of overweight and obese adults with visceral obesity. Elsewhere, it has been demonstrated that metabolically obese and normal-weight Japanese adults have increased levels of visceral fat, associated with elevated TG and insulin resistance [7], and unfavorable lipid profiles have been reported in overweight and obese Chinese adults with high levels of visceral fat [24].

In the present study, the differences in lipid profiles were more pronounced in women. Women with normal BMI and visceral obesity had 11%, 41%, and 17% higher TC, TG, and LDL, respectively, than normal-weight women without visceral obesity. Similarly, overweight/obese women with visceral obesity had 4%, 31%, and 5% higher TC, TG, and LDL, respectively, than their counterparts without visceral obesity. In both groups, women with visceral obesity were older than those without visceral obesity, suggesting the influences of increasing age and menopause. In agreement, the AUC and ROC analysis found that, after 50 years, women were more likely to accumulate excess visceral fat. Previous studies have shown that adipose tissue is established in a large organ that contains two parenchymal components: white (WAT) and brown adipose tissue (BAT), which are distributed in the subcutaneous and visceral compartments. BAT can be burned to generate energy, while WAT mainly stores energy. The relative amounts and locations of WAT and BAT depend on factors such as age, sex, and so on [9]. Another study showed that menopause and consequential estrogen deficiency are associated with metabolic alterations that lead to a rapid increase in fat mass and a redistribution of fat from peripheral regions to the abdomen. Our findings indicate that Chinese women over the age of 50 years are at an increased risk of visceral obesity and an adverse lipid profile, which may increase the risk of the development of metabolic and cardiovascular disease.

Previous studies have attempted to identify anthropometric phenotypes for an increased risk of visceral obesity and metabolic syndrome in normal-weight individuals [21,25]. In the current study, we identified a higher risk of visceral obesity in Chinese adults at the upper threshold of the normal BMI category in both sexes. For men, the cutoff was 22.5 kg/m^2^. Having a BMI over this threshold increased the risk of visceral obesity by 4.4-fold. For women, the cutoff was 22.3 kg/m^2^, and the elevated risk was 5.2-fold greater. These cutoff points are lower than the BMI threshold for overweight in the Chinese population (24 kg/m^2^) [26]. We did not find value in the measurement of BSA in either sex, and the BMI cutoffs were more effective in identifying individuals with visceral obesity. One possible explanation for this is that individuals with the same BSA can have a large vertical area or a large horizontal area. As such, this makes it possible for someone with a high BSA to be very tall and thin without visceral obesity.

The assessment of visceral obesity is clinically important, and although CT imaging is recognized as a gold standard, it is not always readily accessible, as this method is costly and involves ionizing radiation. In the absence of imaging, the waist circumference and waist-to-height ratio have been used as proxy measures of visceral fat. Compared to BMI, the waist circumference has been shown to be more strongly predictive of visceral obesity measured by magnetic resonance imaging in Chinese adults [27]. Previous studies have also reported that the Chinese visceral adiposity index (CVAI) is a reliable indicator of type-2 diabetes and cardiometabolic risk [28]. The CVAI algorithm incorporates age, BMI, waist circumference, serum triglyceride, and high-density lipoprotein-C.

There are several considerations to account for when interpreting the findings of the current study. Firstly, it is not possible to generalize our findings globally, given that the study samples were recruited mainly from one city in China (Nanjing). There may be differences in lifestyle and culture across different cities, which should be explored in larger population-based studies. However, the high mobility of the populations in Chinese cities and the different living habits of different age groups afford our data a clear, general significance. Secondly, we did not include measurements of the waist circumference, which would have enabled an investigation of the agreement between this proxy measure of visceral adiposity and visceral fat measured by CT. Finally, the retrospective design of the present study may limit the associations and relationships that can be identified, and these should be followed up in a further study.

## 5. Conclusions

In this cohort of Chinese adults, visceral obesity was more common in men than in women with normal BMI and was associated with an adverse lipid profile similar to those of overweight/obese adults with visceral obesity. The risk factors for visceral obesity in this subpopulation were an age of ≥ 50 years or a BMI ≥ 22.3 kg/m^2^ in women and a BMI ≥ 22.5 kg/m^2^ in men. It is recommended that Chinese adults who fall into these categories lose weight, eat healthier, or even receive clinical treatment for visceral adiposity and cardiometabolic disease.

## Figures and Tables

**Figure 1 diagnostics-12-02522-f001:**
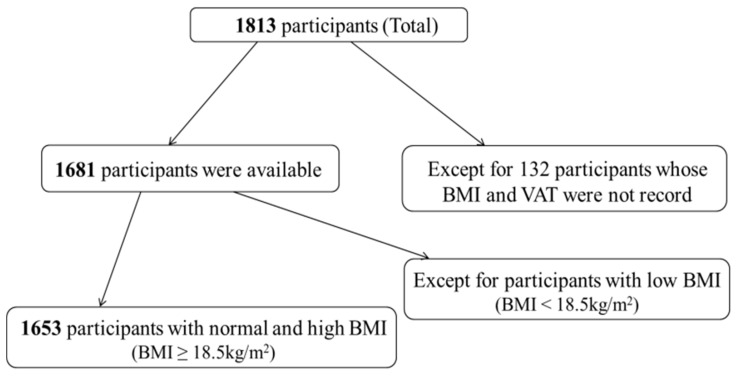
Schematic diagram of participant grouping. A total of 1813 Chinese adults received health assessments and were recruited for the study. From this number, 132 participants were excluded due to invalid BMI or visceral fat data, and 28 were excluded because their BMI was <18.5 kg/m^2^. BMI, body mass index; VAT, visceral adipose tissue.

**Figure 2 diagnostics-12-02522-f002:**
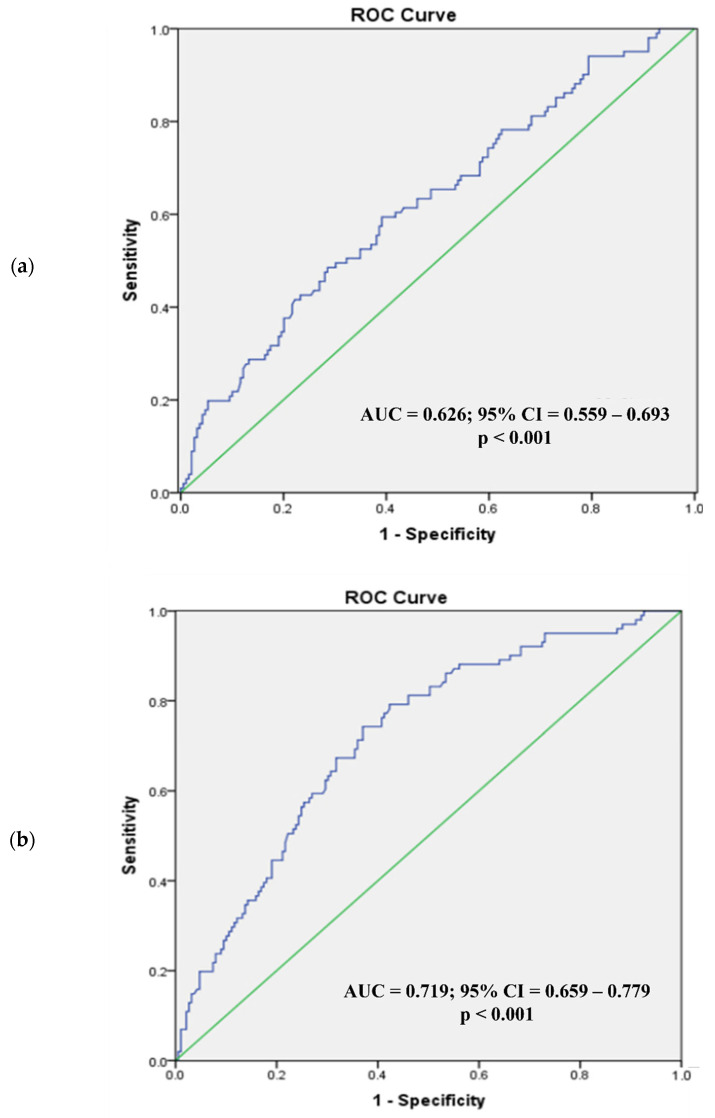
Receiver operating characteristic analysis of men with normal BMI (body surface area (**a**), body mass index (**b**)). In normal-weight men, the AUC for BSA (**a**) was 0.626 (95% CI = 0.559–0.693; *p* < 0.001) and for BMI (**b**) 0.719 (95% CI = 0.659–0.779; *p* < 0.001). ROC, receiver operating characteristic; AUC, area under the curve; CI, confidence interval.

**Figure 3 diagnostics-12-02522-f003:**
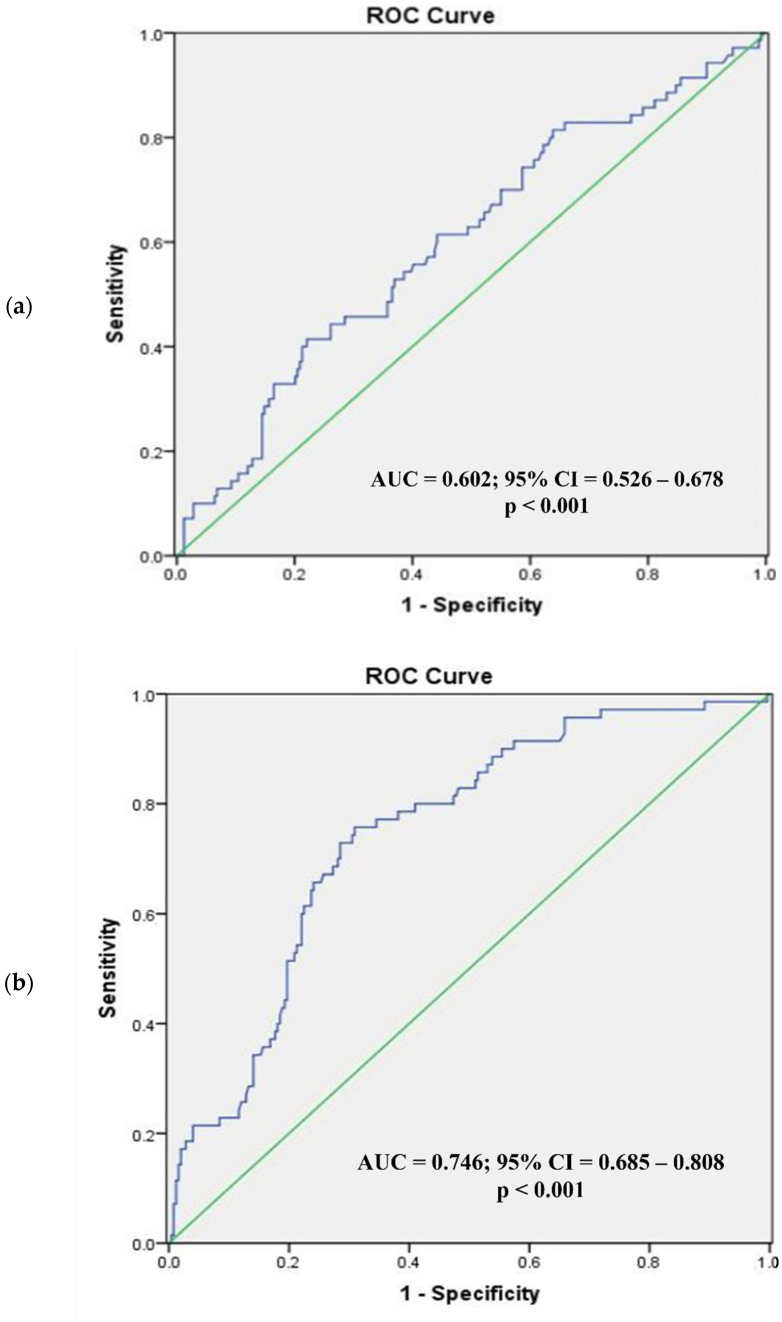
Receiver operating characteristic analysis of women with normal BMI (body surface area (**a**), body mass index (**b**)). In normal-weight women, the AUC for BSA (**a**) was 0.602 (95% CI = 0.526–0.678; *p* < 0.001) and for BMI (**b**) 0.746 (95% CI = 0.685–0.808; *p* < 0.001). ROC, receiver operating characteristic; AUC, area under the curve; CI, confidence interval.

**Figure 4 diagnostics-12-02522-f004:**
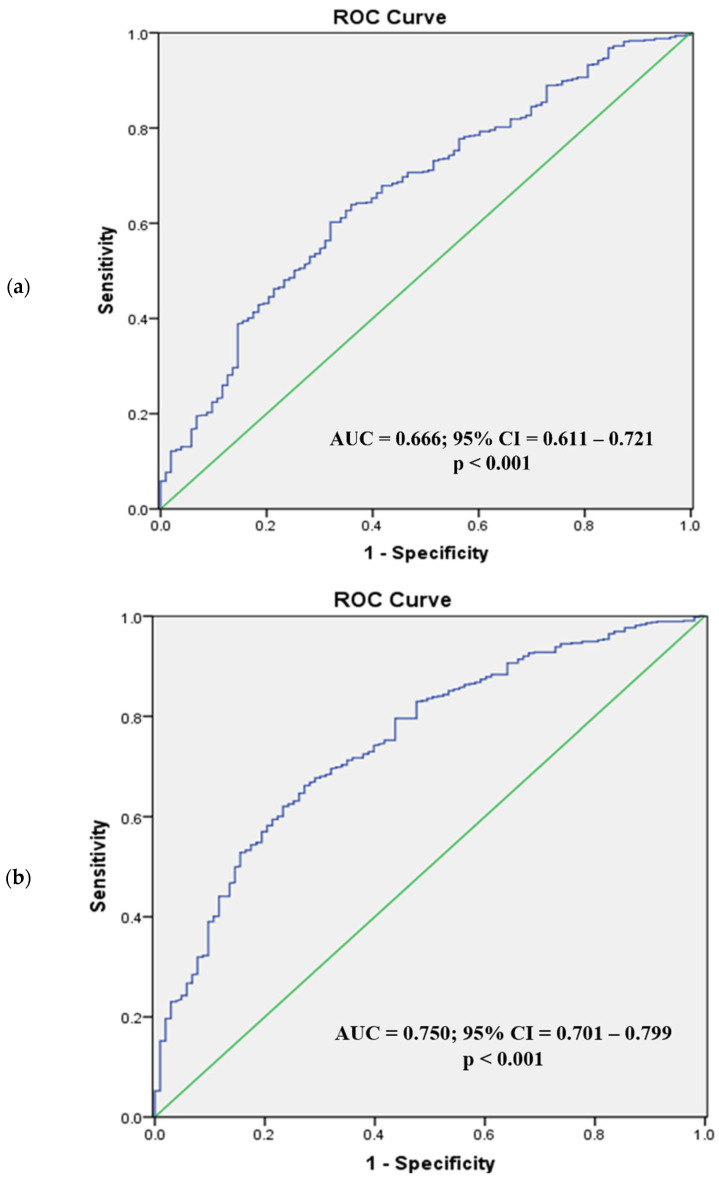
Receiver operating characteristic analysis of men with high BMI (body surface area (**a**), body mass index (**b**)). For men with high BMI, the AUC for BSA (**a**) was 0.666 (95% CI = 0.611–0.721; *p* < 0.001) and for BMI (**b**) 0.750 (95% CI = 0.701–0.799; *p* < 0.001). ROC, receiver operating characteristic; AUC, area under the curve; CI, confidence interval.

**Figure 5 diagnostics-12-02522-f005:**
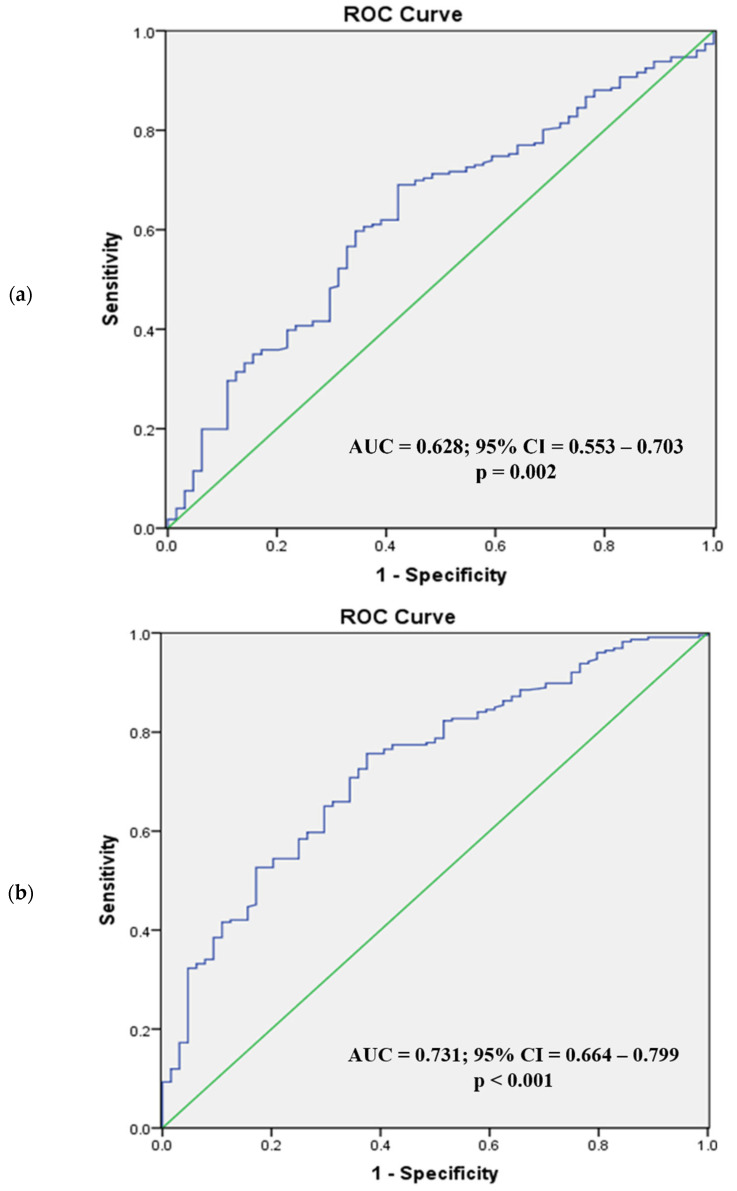
Receiver operating characteristic analysis of women with high BMI (body surface area (**a**), body mass index (**b**)). In high BMI women, the AUC for BSA (**a**) was 0.628 (95% CI = 0.553–0.703; *p* = 0.002) and for BMI (**b**) 0.731 (95% CI = 0.664–0.799; *p* < 0.001). ROC, receiver operating characteristic; AUC, area under the curve; CI, confidence interval.

**Table 1 diagnostics-12-02522-t001:** Descriptive characteristics of Chinese men (*n* = 1044) according to BMI and visceral fat.

		High VAT * (*n* = 101)	Normal VAT (*n* = 189)	Difference (*p*)
Normal BMI (*n* = 290)	VAT area, cm^2^	179.70 ± 25.55	96.48 ± 30.17	-
	Age, y	48.39 ± 11.54	46.32 ± 11.02	0.14
	Weight, kg	66.73 ± 5.12	63.45 ± 5.54	<0.001
	Height, cm	170.78 ± 5.77	170.13 ± 5.53	0.35
	BMI, kg/m^2^	22.86 ± 1.01	21.90 ± 1.37	<0.001
	BSA, m^2^	1.74 ± 0.096	1.70 ± 0.097	<0.001
		**(*n* = 651)**	**(*n* = 103)**	
Overweight and obese (*n* = 754)	VAT area, cm^2^	216.28 ± 48.88	114.16 ± 26.57	-
	Age, y	46.93 ± 10.58	47.24 ± 10.88	0.78
	Weight, kg	80.20 ± 8.83	74.19 ± 7.08	<0.001
	Height, cm	170.56 ± 6.05	169.82 ± 6.90	0.25
	BMI, kg/m^2^	27.54 ± 2.39	25.70 ± 1.55	<0.001
	BSA, m^2^	1.91 ± 0.14	1.83 ± 0.13	<0.001

VAT = visceral adipose tissue; BMI = body mass index; BSA = body surface area. * High VAT (men ≥ 142 cm^2^).

**Table 2 diagnostics-12-02522-t002:** Descriptive characteristics of Chinese women (*n* = 609) according to BMI and visceral fat.

		High VAT * (*n* = 70)	Normal VAT (*n* = 249)	Difference (*p*)
Normal BMI (*n* = 319)	VAT area, cm^2^	144 ± 24.3	73.5 ± 24.6	-
	Age, y	55.6 ± 10.2	44.9 ± 11.2	<0.001
	Weight, kg	56.9 ± 4.81	54.4 ± 4.75	<0.001
	Height, cm	158 ± 5.29	159 ± 5.61	0.50
	BMI, kg/m^2^	22.7 ± 1.04	21.5 ± 1.39	<0.001
	BSA, m^2^	1.54 ± 0.09	1.51 ± 0.09	0.013
		**(*n* = 226)**	**(*n* = 64)**	
Overweight and obese (*n* = 290)	VAT area, cm^2^	161 ± 35.5	96.1 ± 15.1	-
	Age, y	53.2 ± 10.9	48.3 ± 9.95	0.002
	Weight, kg	66.6 ± 7.25	63 ± 5.41	<0.001
	Height, cm	157 ± 5.50	157 ± 6.21	0.58
	BMI, kg/m^2^	27.1 ± 2.41	25.5 ± 1.37	<0.001
	BSA, m^2^	1.66 ± 0.12	1.61 ± 0.10	0.007

VAT = visceral adipose tissue; BMI = body mass index; BSA = body surface area. * High VAT (women ≥ 115 cm^2^).

**Table 3 diagnostics-12-02522-t003:** Multivariable analysis of risk factors for visceral obesity in Chinese adults with normal BMI (*n* = 609).

Estimated Factors	Odds Ratio	95% Confidence Interval	*p*
Men (*n* = 290)			
Age (≥50 vs. <50 years)	1.62	0.93–2.81	0.09
BSA (>1.72 vs. ≤1.72 m^2^)	1.64	0.94–2.87	0.08
BMI (>22.5 vs. ≤22.5 kg/m^2^)	4.41	2.52–7.69	<0.001
Women (*n* = 319)			
Age (≥50 vs. <50 Years)	5.43	2.82–10.5	<0.001
BSA (>1.57 vs. ≤1.57 m^2^)	2.41	1.23–4.71	0.07
BMI (>22.3 vs. ≤22.3 kg/m^2^)	5.18	2.71–9.89	<0.001

BMI = body mass index; BSA = body surface area.

**Table 4 diagnostics-12-02522-t004:** Multivariable analysis of risk factors for visceral obesity in Chinese adults with high BMI (*n* = 1044).

Estimated Factors	Odds Ratio	95% Confidence Interval	*p*
Men (*n* = 754)			
BSA (>1.87 vs. ≤1.87 m^2^)	1.84	1.14–2.96	0.012
BMI (>26.3 vs.≤26.3 kg/m^2^)	4.26	2.61–6.96	<0.001
Women (*n* = 290)			
Age (≥50 vs. <50 Years)	3.77	1.94–7.31	<0.001
BSA (>1.62 vs. ≤1.62 m^2^)	2.99	1.53–5.85	0.001
BMI (>25.3 vs. ≤25.3 kg/m^2^)	3.99	2.13–7.46	<0.001

BMI = body mass index; BSA = body surface area.

**Table 5 diagnostics-12-02522-t005:** Lipid profiles of Chinese men (*n* = 1044) according to BMI and visceral fat.

		High VAT *n* = 101	Normal VAT *n* = 189	Difference *p*
Normal BMI (*n* = 290)	TC	5.04 ± 0.83	4.76 ± 0.84	0.007
	TG	1.80 ± 1.05	1.42 ± 1.42	0.02
	LDL/HDL	2.54 ± 0.78	2.15 ± 0.86	<0.001
	LDL	3.11 ± 0.68	2.84 ± 0.72	0.002
	HDL	1.29 ± 0.30	1.43 ± 0.39	0.001
		***n* = 651**	***n* = 103**	
Overweight and obese (*n* = 754)	TC	5.02 ± 0.92	4.89 ± 1.02	0.18
	TG	2.37 ± 2.22	1.70 ± 1.39	<0.001
	LDL/HDL	2.71 ± 0.91	2.41 ± 0.87	0.002
	LDL	3.02 ± 0.80	3.00 ± 0.83	0.81
	HDL	1.17 ± 0.29	1.31 ± 0.31	<0.001

BMI = body mass index, VAT = visceral adipose tissue, TC = total cholesterol, TG = total glucose, LDL = low density lipoprotein, HDL = high density lipoprotein.

**Table 6 diagnostics-12-02522-t006:** Lipid profiles of Chinese women (*n* = 609) according to BMI and visceral fat.

		High VAT *n* = 70	Low VAT *n* = 249	Difference *p*
Normal BMI (*n* = 319)	TC	5.32 ± 1.06	4.75 ± 0.85	<0.001
	TG	1.75 ± 1.54	1.03 ± 0.55	<0.001
	LDL/HDL	3.41 ± 9.00	1.71 ± 0.67	0.12
	LDL	3.24 ± 0.89	2.71 ± 0.75	<0.001
	HDL	1.44 ± 0.37	1.68 ± 0.39	<0.001
		***n* = 226**	***n* = 64**	
Overweight and obese (*n* = 290)	TC	5.12 ± 0.99	4.93 ± 0.89	0.16
	TG	1.84 ± 1.84	1.23 ± 0.74	<0.001
	LDL/HDL	2.30 ± 0.70	1.99 ± 0.73	0.002
	LDL	3.06 ± 0.78	2.95 ± 0.76	0.31
	HDL	1.39 ± 0.36	1.57 ± 0.40	0.001

BMI = body mass index, VAT = visceral adipose tissue, TC = total cholesterol, TG = total glucose, LDL = low density lipoprotein, HDL = high density lipoprotein.

## Data Availability

The datasets used and analyzed in the current study are available from the corresponding author upon reasonable request.

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
