# Peer review of "Visceral Obesity and Lipid Profiles in Chinese Adults with Normal and High Body Mass Index"

_diagnostics, 2022, doi:10.3390/diagnostics12102522_

Round 1
Reviewer 1 Report
This study evaluated the prevalence and phenotypic predictors of visceral obesity in Chinese adults according to different BMI ranges, and associations with some biochemical blood parameters.
The manuscript, raises the current issues, although suffers many structural and perhaps conceptual defects.
Please find below my comments:
Line 19: CT - this abbreviation should be explained
Line 23: BMI values for normal and high categories should be mentioned in this line
Line 27: I suggest changing the word „person” (or persons) to „patients”, „participants” etc.
Line 28: It is a duplication of lines 26 and 27. I suggest changing this sentence e.g.: „The BMI cut-off points are lower than officially accepted”
Line 36-38: These lines should also contain percentage values indicating obesity in women and men according to the WHO
Lines 38-41: This sentence may be misleading. As a rule, BMI is not used to assess the cardio-metabolic risk. There are some anthropometric indices frequently showed a greater diagnostic ability of visceral or sarcopenic obesity and comorbidities, compared to BMI (e.g. ABSI, BRI, VAI, CI, and others). BMI is an indicator used for the diagnosis of obesity. It also can be mentioned that BMI, although useful for population studies, has significant limitations when applied to individuals.
Line 53: „whole body visceral adiposity” unclear sentence.
Line 71: The "Material and methods" section lacks a lot of important information: Where the examination was conducted (university, clinic, hospital, etc.?) In what region of China?
Moreover:
- The number of women and men, the average age of the whole group, average BMI value
- What methods were used to recruit participants?
- Was there an age limit? Contraindications to CT should be listed as excluding factors
This section also should be mentioned that all participants gave their informed consent for inclusion before the study.
Line 74: The abbreviation VAT should be explained. Is there a difference between VAT and VF?
Line 84: How many people were performing measurements? Were patients fully dressed during body weight measurement?
Line 92: How many radiologists?
Line 92: CT - the full name of the method should be used
Lines 139 and 142: What about Table 1a (men characteristic?). Table 1b is duplicated.
Lines: 231-233: Unclear results presentation. A number of what?
The motivation for this study is not well chosen. A mere presented the prevalence and predictors of visceral obesity in adults with normal and high BMI, was previously widely describe and do not seem provide scientific advancements. Moreover, the manuscript requires a thorough language editing.
Reviewer 2 Report
It is a useful article to photograph the current situation.
I suggest citing the studies of Prof. Saverio Cinti on the adipose organ. In mammals, the adipose tissues are contained in a multi-depot organ: the adipose organ. It consists of several subcutaneous and visceral depots. Some areas of these depots are brown and correspond to brown adipose tissue, while many are white and correspond to white adipose tissue. The organ is rich of vessels and parenchymal nerve fibers, but their density is higher in the brown areas. White areas contain a variable amount of brown adipocytes and their number varies with age, strain and environmental condition.
Round 2
Reviewer 1 Report
Much thanks to the authors for incorporating majorrecommendations from the review, with the paper better suitable for publication.